# Potential of the Stromal Matricellular Protein Periostin as a Biomarker to Improve Risk Assessment in Prostate Cancer

**DOI:** 10.3390/ijms23147987

**Published:** 2022-07-20

**Authors:** Valentina Doldi, Mara Lecchi, Silva Ljevar, Maurizio Colecchia, Elisa Campi, Giovanni Centonze, Cristina Marenghi, Tiziana Rancati, Rosalba Miceli, Paolo Verderio, Riccardo Valdagni, Paolo Gandellini, Nadia Zaffaroni

**Affiliations:** 1Molecular Pharmacology Unit, Department of Applied Research and Technological Development, Fondazione IRCCS Istituto Nazionale dei Tumori, 20133 Milan, Italy; valentina.doldi@istitutotumori.mi.it (V.D.); nadia.zaffaroni@istitutotumori.mi.it (N.Z.); 2Bioinformatics and Biostatistics Unit, Department of Applied Research and Technological Development, Fondazione IRCSS Istituto Nazionale dei Tumori, 20133 Milan, Italy; mara.lecchi@istitutotumori.mi.it (M.L.); paolo.verderio@istitutotumori.mi.it (P.V.); 3Clinical Epidemiology and Trial Organization, Department of Applied Research and Technological Development, Fondazione IRCCS Istituto Nazionale dei Tumori, 20133 Milan, Italy; silva.ljevar@istitutotumori.mi.it (S.L.); rosalba.miceli@istitutotumori.mi.it (R.M.); 4Department of Pathology and Laboratory Medicine, Fondazione IRCCS Istituto Nazionale dei Tumori, 20133 Milan, Italy; colecchia.maurizio@hsr.it (M.C.); elisa.campi@istitutotumori.mi.it (E.C.); giovanni.centonze@istitutotumori.mi.it (G.C.); 5Prostate Cancer Program, Fondazione IRCCS Istituto Nazionale dei Tumori, 20133 Milan, Italy; cristina.marenghi@istitutotumori.mi.it (C.M.); tiziana.rancati@istitutotumori.mi.it (T.R.); riccardo.valdagni@istitutotumori.mi.it (R.V.); 6Division of Radiation Oncology, Fondazione IRCCS Istituto Nazionale dei Tumori, 20133 Milan, Italy; 7Department of Oncology and Hemato-Oncology, University of Milan, 20133 Milan, Italy; 8Department of Biosciences, University of Milan, 20133 Milan, Italy

**Keywords:** prostate cancer, circulating biomarkers, periostin, sparc, active surveillance

## Abstract

Prostate cancer (PCa) ranges from indolent to aggressive tumors that may rapidly progress and metastasize. The switch to aggressive PCa is fostered by reactive stroma infiltrating tumor foci. Therefore, reactive stroma-based biomarkers may potentially improve the early detection of aggressive PCa, ameliorating disease classification. Gene expression profiles of PCa reactive fibroblasts highlighted the up-regulation of genes related to stroma deposition, including periostin and sparc. Here, the potential of periostin as a stromal biomarker has been investigated on PCa prostatectomies by immunohistochemistry. Moreover, circulating levels of periostin and sparc have been assessed in a low-risk PCa patient cohort enrolled in active surveillance (AS) by ELISA. We found that periostin is mainly expressed in the peritumoral stroma of prostatectomies, and its stromal expression correlates with PCa grade and aggressive disease features, such as the cribriform growth. Moreover, stromal periostin staining is associated with a shorter biochemical recurrence-free survival of PCa patients. Interestingly, the integration of periostin and sparc circulating levels into a model based on standard clinico-pathological variables improves its performance in predicting disease reclassification of AS patients. In this study, we provide the first evidence that circulating molecular biomarkers of PCa stroma may refine risk assessment and predict the reclassification of AS patients.

## 1. Introduction

Prostate cancer (PCa) ranges from indolent tumors, which never develop into clinically relevant diseases, to aggressive and invasive diseases that may rapidly progress and metastasize. The shift from indolent to aggressive PCa is fostered by multiple biological events, including the progressive deposition of a reactive stroma infiltrating tumor foci [1]. Stroma mostly consists of fibroblasts and structural extracellular matrix (ECM) proteins, including collagens and laminins. In normal conditions, by releasing paracrine growth factors and under the androgen hormone influence, stroma cells nourish and orchestrate the physiologic development of the prostate gland, ensuing epithelial morphogenesis [2]. Analogously, prostate stroma components substantially cooperate in carcinogenic development by establishing a tumor-supportive environment [3].

Within this tumor-supportive stroma, the physiological components, such as fibroblasts, are mainly replaced by activated fibroblasts and myofibroblasts, the so-called cancer-associated fibroblasts (CAF), which are responsible for the synthesis, remodeling, and deposition of tumor desmoplastic stroma. Dynamic changes also occur in ECM composition where the structural mixture of collagens, proteoglycans, and matricellular proteins, such as SPARC, thrombospondin-1, and hyaluronan, are profoundly remodeled, and laminins are replaced by tenascin-C [3,4,5]. By acquiring desmoplastic features, PCa reactive stroma can influence the acquisition of an aggressive phenotype and define the nature of the disease. For instance, experimental evidence revealed that ECM derived from prostate reactive stroma could promote proliferation, survival, and motility of PCa cells. In this context, we have previously found that CAF activation in PCa is guided by pivotal cancer-derived factors, such as Transforming Growth Factor β (TGF-β) and Interleukin 6 (IL-6), which can both act by creating an inflammatory and hypoxic stroma microenvironment suitable for tumor progression and spreading [6,7,8].

PCa is a highly heterogeneous disease in terms of clinical presentation. As for many tumor conditions, the early and univocal identification of aggressive disease is mandatory for treatment decision-making and to ultimately improve patient outcomes. Therefore, several efforts have been made to define the key biological signatures sustaining PCa progression with the prospective to introduce innovative tools for the early detection of aggressive PCa, especially for patients enrolled in Active Surveillance (AS) programs. AS has evolved as an alternative to radical treatment for patients with low-risk/very low-risk, potentially indolent PCa. Conventionally, patients harboring tumors classified as Gleason Score (GS) = 3 + 3 or Prognostic Grade Group 1 (PGG1) are suitable for AS, provided that they have other favorable clinico-pathological features [9,10]. During the program, patients are strictly monitored through clinical examination, serial measurements of PSA levels, and scheduled prostatic repeated biopsies; radical treatments are avoided or deferred until evidence of more aggressive PCa, without missing the curability window [11]. Though AS has been shown to be a safe approach in the medium-long term, it is worth reporting that about 20% of patients discontinue AS prematurely due to disease up-grading at repeated biopsies (i.e., finding of GS > 3 + 3 or PGG > 1 cancer foci) [12]. Early dropout mainly reflects misclassification of the disease during the initial diagnostic procedures due to incomplete biopsy sampling but could also reflect the emergence of a more aggressive cancer. This phenomenon highlights the urgent need for novel biomarkers for accurate identification of patients that, though being eligible for AS based on current clinico-pathological criteria, instead harbor occult (or will develop) high-risk tumors and should be instead addressed to radical treatments. In this scenario, alterations of PCa stroma found to be associated with an aggressive disease might be potentially used as novel molecular biomarkers for refining the disease state at diagnosis.

High periostin expression in PCa specimens examined in toto has been associated with shorter biochemical recurrence (BCR)-free survival in patients [13]. Interestingly, Özdemir and colleagues recently showed that primary and bone metastatic PCa displayed strong periostin immunoreactivity in myofibroblasts over the entire tumor stroma, while cancer cells were negative [14].However, the role of stromal periostin and the contribution of its deregulated expression in PCa has not yet been fully elucidated. In this work, we identified a panel of up-modulated genes in reactive fibroblasts, encoding for the ECM proteins periostin and sparc. Here, we report that PCa periostin is mainly expressed in the peritumoral stroma and not in tumor cells. In addition, its stromal expression correlates with PCa grade and BCR in PCa patients. Based on the evidence that periostin is also released in the extracellular space and plasma, we show that the integration of circulating periostin and sparc levels with a panel of clinico-pathological variables in AS patients improves the performance of conventional parameters to predict disease upgrading. Overall, in this study, we established that molecular features characteristic of PCa stroma are associated with adverse diseases and can potentially improve risk assessment, thus limiting disease misclassification.

## 2. Results

### 2.1. An ECM-Related Signature Is Up-Modulated in CAFs Surrounding PCa Foci

By analyzing the gene expression profile of patient-derived prostate CAFs and normal prostate fibroblasts (NPFs) activated in vitro with IL-6 or TGF-β (data from Doldi et al. [8], deposited at GEO DataSets as GSE76260), we identified a panel of ECM genes (POSTN, SPARC, COMP, COL11A1, and COL10A1) up-modulated in at least one type of reactive samples (CAFs or in vitro activated fibroblasts) compared to NPFs, suggesting that the composition of ECM in the peritumoral stroma of PCa may be distinct from that of normal tissue. Among these ECM genes, POSTN resulted as the most up-regulated and concomitantly enriched in all types of reactive fibroblasts (FC = 2.18 in CAFs vs. NPFs, *p*-value = 0.0076; FC = 2.47 in NPF-IL6 vs. NPFs, *p*-value = 0.0012; FC = 2.14 in NPF-TGF-β vs. NPFs, *p*-value = 0.0002).

To investigate the significance of stromal POSTN up-modulation, we primarily evaluated the expression levels of periostin in an independent series of paired CAFs and NPFs established from three prostatectomy samples collected in our Institute. qRT-PCR and Western blotting analysis confirmed that both periostin mRNA and protein levels were up-modulated in CAFs with respect to matched NPFs (Figure 1a,b), thus corroborating our initial observation. Interestingly, qRT-PCR analysis on PCa cell lines (DU145 and PC-3) indicated that POSTN expression levels were higher in CAFs than in PCa cells (Figure 1a).

### 2.2. Stromal Periostin Is Associated with PCa Tumor Grade and Biochemical Relapse

To determine whether periostin could be associated with PCa aggressiveness, tissue sections from radical prostatectomy specimens obtained from 116 PCa patients (Appendix A) were used to evaluate periostin staining by IHC. Consistent with the results obtained in the cell models, periostin was found to be prevalently expressed in the peritumoral stroma rather than in tumor cells (Figure 2a), suggesting that periostin over-expression found in prostate tumors is mainly due to its specific up-modulation in the peritumoral reactive stroma [13].

Interestingly, non-neoplastic areas almost invariably showed undetectable or weak periostin staining. PGG1/GS = 3 + 3 tumors displayed predominantly undetectable (39.5%) or weak (44.7%) peritumoral periostin staining, suggesting that low-grade PCa is mostly surrounded by non-reactive stroma (Figure 2b). Differently, clinically significant PCa—classified as PGG2 (GS = 3 + 4) and PGG ≥ 3 (GS ≥ 4 + 3)—showed a high percentage of medium/strong peritumoral periostin staining, which increased progressively with the increase intumor grade, resulting associated with the aggressiveness of the disease (Chi-Square *p*-value < 0.0001). Specifically, 25.0% of PGG2 tumors exhibited strong stromal periostin staining. For PGG3 and PGG > 3 tumors, strong staining reached 55.0% and 72.2% of the cases, respectively (Figure 2b). In addition, we observed a borderline association (Fisher exact *p*-value = 0.0516) of stromal periostin staining and the presence of cribriform morphology in pattern 4 foci in the sub-cohort of GS = 3 + 4/4 + 3 (PGG2/3) tumors (*n* = 60, of which 14 with cribriform features and 46 without). As shown in Figure 2c, we found that 100% of tumors with cribriform pattern 4 foci displayed a high (medium/strong) periostin stroma staining. Conversely, althoughthe high periostin stroma score was found to be relevant for PGG2/3 tumors, 73.9% of the tumors characterized by non-cribriform morphology was classified as high (medium/strong) score and 26.1% as low (undetectable/weak). Full images are reported in Appendix A. These findings suggest a possible association between peritumoral reactive PCa stroma and the emergence of aggressive features, as is cribriform tumor growth.

At a median follow-up of 84 months (range, 65–104 months), the probability of BCR-free survival was 0.78 (95% CI, 0.70–0.85) among the 116 PCa patients, of which 31 experienced a biochemical failure. Univariate Cox analysis showed that periostin, resection margins, extraprostatic extension, prognostic grade group and seminal vesicle invasion were associated with an increased risk of experiencing a biochemical failure, as reported in Table 1. Among the aforementioned variables, periostin proved to be the most predictive (HR: 6.96; *p*-value = 0.0015). The log-rank test referred to as periostin showed statistical significance in terms of BCR-free survival (Figure 2d). In the final multivariate Cox model, periostin (HR: 5.75; *p*-value = 0.0044) and resection margins (HR: 3.47; *p*-value = 0.0016) maintained their prognostic value in terms of BCR-free survival. The overall capacity of the variables to predict BCR-free survival in the final model was overall satisfactory (C = 0.74). When we considered the contribution of each variable (Appendix A), periostin and resection margins had very comparable predictive capabilities.

### 2.3. Circulating Periostin Levels Display Modest Capability to Detect High-Grade Tumors in AS Patients

Gleason score represents one of the strongest predictors of clinical outcome in PCa patients. Considering the association observed between stromal periostin staining and tumor grade (Figure 2b), we envisioned the possibility that circulating periostin may be a potential marker to identify high-grade PCa carriers. To preliminarily test whether stromal periostin could be released in the extracellular fluids, its protein levels were assessed in the condition medium (CM) of patient-derived CAFs and NPFs. Western blotting analysis showed that periostin is actually secreted in the extracellular fluids and more abundantly in the CM from CAFs with respect to NPFs (Figure 3a).

Then, we explored the possibility of detecting periostin in PCa patients’ plasma. Western blotting performed on a small set of patients showed that periostin protein is indeed detectable in circulation, especially in patients with clinically relevant diseases (Pt5–6, Figure 3a). In plasma samples collected from patients at inclusion in AS protocols, circulating periostin was apparently higher in patients who later discontinued the program (up-grading group, Pt3–4, Figure 3a) as compared to truly indolent patients (indolent group, Pt1–2, Figure 3a). To validate this hypothesis, the analysis was extended to plasma samples collected from 100 AS patients and made quantitative using ELISA-based assessment. In this set of patients, circulating periostin levels appeared as not statistically different between the indolent and up-grading group (Figure 3b). Logistic models based on the modulation of periostin as a continuous variable showed that circulating levels of periostin displayed only a modest capability to discriminate between the indolent and the up-grading group (AUC = 0.547; 95% CI: 0.431–0.661) (Appendix A). In contrast, circulating periostin was confirmed to be significantly increased in the plasma of 21 patients with the clinically relevant disease (5-fold more than the AS group), indicating that a high periostin secretion is associated with aggressive tumors (Figure 3b).

### 2.4. Index Score Model including Circulating Periostin and Sparc Improves the Predictive Value of Clinico-Pathological Variables

To define predictive models based on the combination of multiple variables, circulating levels of sparc have been assessed in our cohort of 100 AS patients. Similar to periostin, circulating sparc per se was not differentially expressed between indolent and up-grading groups (Figure 3c). When sparc was modeled as a continuous variable, its capability to discriminate between the indolent and the up-grading group was modest (AUC = 0.599; 95% CI: 0.487–0.708) (Appendix A). A logistic model obtained by integrating periostin and sparc circulating levels slightly improved the risk predictive performance (AUC: = 0.616; 95% CI: 0.505–0.726) (Appendix A). Interestingly, the discriminatory capability of the model was instead improved when circulating sparc and periostin were analyzed as categorical variables and implemented as index score within a clinico-pathological model (Figure 3d). Specifically, the Harrell’s C index of the model was enhanced from 0.739 to 0.791 (with superimposable improvement of model-derived ROC-AUCs) when periostin and sparc were added as categorical values to selected clinico-pathological variables (Figure 3d).

## 3. Discussion

In this study, we investigated the potential of stroma-derived biomarkers as novel tools to improve the risk assessment of PCa patients. By analyzing the gene expression profile of patient-derived CAFs, we identified a signature of ECM proteins over-expressed in PCa peritumoral stroma. Among them, we found that stromal periostin expression was significantly associated with aggressive disease and adverse patient outcomes. Moreover, for the first time, we provided evidence that circulating stromal biomarkers may improve the performance of standard clinico-pathological variables in predicting disease reclassification for AS patients.

Increasing efforts are needed to refine the common risk assessment tools and ensure the early and univocal identification of patients with truly indolent vs. clinically relevant PCa. In this regard, signals from the tumor microenvironment could potentially add fundamental information and improve the accuracy of prognostication based on clinico-pathological parameters only. Morphological and functional changes have indeed been observed in PCa stroma, including altered ECM synthesis and switch of cellular components from normal fibroblasts to CAFs and myofibroblasts [3,15,16]. Such desmoplastic tumor microenvironment frequently accompanies neoplastic lesions and reflects the disease course, ultimately influencing the response to therapies [17,18]. An unexpected result recently emerged from the transcriptional analysis of PCa specimens obtained via radical prostatectomy has confirmed the crucial role of the stroma in tumor progression [19]. Specifically, the most prominent differences in terms of gene expression between high-grade and low-grade tumor foci were found in the infiltrating stroma rather than in the epithelial compartment. In addition, PCa stroma microdissected far away from tumoral foci significantly differs from that derived from non-neoplastic prostate tissues, thus highlighting that alterations in the microenvironment may even precede epithelial cell transformation [19]. Given the pivotal role of peritumoral stroma, increasing efforts have been made to define stromal prognostic signatures. For instance, pathways related to tissue remodeling, inflammation, and cell differentiation were found to be deregulated in the peritumoral stroma and significantly associated with disease progression in several studies [20,21].

Accordingly, our explorative analysis highlighted that PCa reactive stroma is characterized by the up-modulation of genes involved in ECM remodeling, including periostin, sparc, collagens, and COMP. Generally absent in most adult tissues [22], periostin was found to be over-expressed in clinical specimens and significantly associated with aggressive disease both in primary and advanced PCa [23]. Little is known about the role of periostin in establishing a dysfunctional tumor-supportive microenvironment. Functional studies showed that periostin seems to be involved in shaping the tumor microenvironment through the regulation of multiple processes, including the maintenance of the stem-cell niche, the promotion of angiogenesis, the enhancement of tumor cell proliferation, and the formation of the osteoblastic PCa bone metastatic niche [14]. In addition, periostin can foster tumor invasion and metastasis by inducing epithelial-to-mesenchymal transition in cancer cells [24,25].

The most relevant studies investigating periostin in PCa, which were mainly focused on assessing the correlation between periostin expression, tumor grade, and patient outcome, showed contradictory results and missed to clearly define the identity of periostin-expressing cells within the tumor mass. In this regard, Tsunoda et al. found an increased expression of periostin in tumor cells of early-stage PCa and in the peritumoral stroma of advanced tumors [26]. Differently, in a large cohort of PCa patients, Tishchler and colleagues reported that epithelial periostin expression significantly correlates with tumor grade and disease stage but that stromal periostin correlates only with the disease stage [13]. More recently, Cattrini’s group demonstrated that total periostin expression (including both epithelial and stromal compartments) was an independent prognostic factor for advanced PCa patients [23]. Globally, our findings are consistent with previous observations showing a significant correlation between stromal periostin expression and clinical features, though, in our hands, PCa cells seemed to express negligible periostin levels as compared to stromal cells, both in vitro and in clinical specimens. We found that stromal periostin per se was associated with high-grade tumors and shorter BCR-free survival in our PCa patient cohort. Specifically, we observed that peritumoral periostin expression was associated with the most advanced grades of the disease, displaying a progressive increase from PGG2 to PGG > 3 tumors. In addition, our results indicated a positive association between peritumoral periostin expression and aggressive disease features, such as the cribriform morphologic pattern. PCa tumors with cribriform architecture are indeed more lethal as compared to non-cribriform counterparts [27].

Especially for AS, where incomplete tumor sampling during diagnostic biopsies represents one of the leading causes of tumor misclassification, differences in tumor stroma between patients with truly indolent tumors and carriers of occult high-grade cancer can be potentially used as biomarkers for risk assessment. Tumor signatures, such as peculiar ECM protein composition or secretion, may indeed represent early signs of aggressive disease, which may help treatment decision-making and, for instance, direct suspected patients to radical treatments. In this perspective, and with the long-term aim to encourage the development of minimally invasive prognostic tools based on liquid biopsy, we assessed the circulating levels of stromal-derived biomarkers in a cohort of AS patients. Our findings highlighted that periostin and sparc are detectable in the blood of AS patients and that their levels, when jointly considered, can ameliorate the discriminative power of models based on clinico-pathological variables only. In light of these data, it is anticipated that risk stratification may be further improved by combining multiple circulating stromal proteins.

Despite the limited predictive potential of circulating periostin alone in the AS cohort, the 5-fold higher levels found in patients with advanced disease are in agreement with the findings by Cattrini and collaborators, who detected periostin transcript in the blood of metastatic PCa patients and found a positive association with clinical data [23]. This evidence would support the use of periostin as a single marker to monitor disease progression in patients subjected to radical treatments or receiving systemic therapy for metastatic disease, an aspect that deserves validation in future studies.

Overall, our findings suggest that molecular features of tumor stroma may represent promising biomarkers to improve the prognostic performance of models based on clinico-pathological variables for PCa patients, including those eligible for AS. However, we acknowledge that, in this setting, the use of innovative circulating biomarkers to refine risk stratification is still at an early stage and that further studies in large prospective series are necessary. In this regard, aspects that need to be addressed are the selection of relevant circulating molecules, their validation as biomarkers as single entities or in combined models and the identification of the clinical settings (AS vs. clinically significant disease) where their use can provide significant benefit.

## 4. Materials and Methods

### 4.1. Patient Cohort and Samples

For immunohistochemistry analysis, formalin-fixed paraffin-embedded (FFPE) specimens were obtained from 116 patients who received radical prostatectomy between 2005 and 2014 at Fondazione IRCCS Istituto Nazionale dei Tumori, Milano (INT). Original grading of study prostatectomies was assigned using the ISUP 2005 guidelines. A recent revision by the study pathologist (M.C.) showed that all GS = 3 + 3 can be attributed to new prognostic grade group 1 (PGG1), GS = 3 + 4 to new PGG2, GS = 4 + 3 to PGG3, according to the Consensus Conference ISUP 2014 criteria. Appendix A describes clinico-pathological characteristics of PCa patients who underwent radical prostatectomy.

For plasma analysis, low-risk PCa patients included in the study were prospectively accrued into AS protocols open to enrollment between 2008 and 2015 at INT. Two sets of AS patients were considered. The first set, referred to as the indolent group, includes patients on AS for at least 5 years (i.e., no evidence of tumor reclassification at any of the repeated biopsies). The second set, referred to as the up-grading group, includes patients who dropped out from AS due to upgrading (i.e., finding of GS > 3 + 3 at a re-biopsy) within 1 year from inclusion. For comparative purposes, plasma samples from a small set of clinically significant PCa patients who underwent radical radiotherapy at INT between 2011 and 2012 were also considered.

Plasma samples from AS patients and clinically significant PCa patients were collected at patient inclusion in AS or at diagnosis, respectively. Appendix A reports the clinico-pathological variables of the study cohorts. All patients gave written informed consent to donate biological material for research purposes and the study was approved by the Institutional Ethical Committee (project approval code: INT 10/11).

### 4.2. Immunohistochemistry Analysis and Evaluation of Periostin Staining

Tissue sections from FFPE prostatectomy specimens were subjected to automated immunohistochemistry on Autostainer Link 48 instrument (Agilent Technologies, Santa Clara, CA, USA) using EnVisionFLEX+ Target Retrieval Solution, a high pH detection system (Dako, Agilent Technologies), according to manufacturer’s instructions. Briefly, tissue sections of 4 μm were deparaffinized in xylene and heat-induced epitope retrieval was performed in EnVisionFLEX+ Target Retrieval Solution, high pH for 30 min at 97 °C. Periostin was identified usingan anti-POSTN antibody (ab14041, Abcam, Cambridge, UK) at the dilution of 1:1000 for 20 min of incubation. Adjacent sections were stained with hematoxylin and eosin for anatomical reference.

The periostin staining score was evaluated by an expert uropathologist (M.C.). M.C. set up a scale ranging from 0 to 3, where 0 represents ‘undetectable staining’, 1 ‘weak staining’, 2 ‘medium staining’, and 3 the highest (strong) stromal staining observed in the tissue sections.

### 4.3. Cell Cultures

Established human PCa cell lines were purchased from American Type Culture Collection (ATCC, Rockville, MD, USA) and cultured in standard conditions. DU145 and PC3 cells were cultured in RPMI-1640 medium (Lonza, Basel, Switzerland) supplemented with 10% FBS (Thermo Fisher Scientific Inc., Waltham, MA, USA). Cell lines were authenticated and periodically monitored by genetic profiling using short tandem repeat analysis (AmpFISTRIdentifiler PCR amplification kit, Thermo Fisher Scientific Inc.). Human normal prostate fibroblasts (NPFs) and cancer-associated fibroblasts (CAFs) were isolated from surgical explantation after patient informed consent in accord with the Ethics Committee of INT (project approval code: INT 154/16). NPFs and CAFs were established from healthy and peritumoral regions, respectively, of the prostate of PCa patients. Tissue samples were obtained aseptically from patients undergoing radical prostatectomy. Tissue were digested overnight in DMEM supplemented with 5% fetal bovine serum and 1× Collagenase-Hyaluronidase Solution (STEM CELL^TM^ Technologies Inc., Vancouver, BC, Canada). Cells suspension was centrifuged at 1500× *g* for 5 min. The resulting fibroblast-rich pellet was suspended and plated in DMEM containing 10% fetal bovine serum and 4 mM L-Glutamine. Fibroblasts were further enriched based on their shorter trypsinization time compared with epithelial/tumor cells. Upon culture establishment (approximately at the 3rd–4th passage), cells were morphologically evaluated (fibroblasts appear as elongated and large cells with thin flat/wavy nuclei), and specific fibroblast markers were assessed by Western blotting and immunofluorescence analysis (i.e., FAP, FSP, CD90, collagens, fibronectin, α-SMA). The lack of epithelial markers (E-cadherin) was also verified. All molecular analyses described in this work were carried out within the 8th passage.

### 4.4. RNA Isolation and RT-qPCR

Total RNA was isolated using QIAzol Lysis Reagent and miRNeasy Mini Kit (QIAGEN, Hilden, Germany) with DNase I digestion (QIAGEN, Hilden, Germany), according to the manufacturer’s instructions. RNA yield and A260/280 ratio were monitored with a NanoDrop ND-2000c spectrophotometer (Thermo Fisher Scientific Inc.). cDNA was synthesized using a high-capacity cDNA Reverse Transcription Kit (Thermo Fisher Scientific Inc.). Quantification of gene expression was assessed by RT-qPCR using No AmpErase TaqMan Universal PCR Master Mix (Thermo Fisher Scientific Inc.) and the specific POSTN TaqMan gene expression assays (Thermo Fisher Scientific Inc.) (TaqMan^®^ assay Hs01566750_m1). For comparative analyses, GAPDH (PN4326317E) was measured as endogenous control. RT-qPCR results were reported as relative quantity (RQ = 2−ddCt) with respect to a calibrator sample, using the comparative Ct (ddCt) method.

### 4.5. Protein Extraction and Western Blotting

For total protein extraction, cells were lysed in lysis buffer (Tris-HCl pH 7.4 10 mM; NaCl 10 mM; Triton X-100%; PMSF 1×; Aprotinin 5 μg/mL; Leupeptin 20 μg/mL) for 30 min on ice and then the supernatant (i.e., proteins) was boiled at 95 °C for 5 min. 20 µg of proteins were equally loaded for all samples in the experiments. For the assessment of proteins secreted by NPFs and CAFs, an equal number of cells was seeded, and then an equal volume of conditioned medium (CM) was collected by clarification for 10 min at 1500× *g* and 5× concentration by using Concentrator Spin 5K MVCO column (Agilent Technologies). 2.5 μL of 5× concentrated CM was loaded for each sample. Proteins from plasma were loaded starting from the same volume of sample (5 μL for each sample). Whole-cell lysates, conditioned media, or plasma samples were resolved in 4–12% precast Bis–Tris sodium dodecyl sulfate-polyacrylamide gel electrophoresis (SDS-PAGE) (NuPAGE, Thermo Fisher Scientific Inc.) for separation and transferred onto Hybond nitrocellulose membranes (GE Healthcare Life Sciences, Buckinghamshire, UK). Filters were blocked for non-specific reactivity by incubation for 1 h at room temperature in 5% skim milk dissolved in 1× PBS-0.1% Tween 20 and overnight at 4 °C probed with the following antibodies: Periostin (ab14014, Abcam) 1:500, Col10a1 (LS-B14811, Lifespan Biosciences, Seattle, WA, USA) 1:500, Comp (NB100-2478, Novus Biologicals, Littleton, CO, USA) 1:2000, Sparc (ON1-1 33-5500, Thermo Fisher Scientific Inc.) 1:1000. Anti-GAPDH antibody (G8795, Sigma-Aldrich) was used as a control for equal protein loading. After three washes with 1× PBS-0.1% Tween 20, filters were incubated with the secondary horseradish peroxidase-conjugated anti-mouse (NA931V, GE Health-care Life Sciences) or anti-rabbit (NA9340V, GE Healthcare Life Sciences) antibodies for 1 h at room temperature. Immunoreactivity was detected by the enhanced chemiluminescence (ECL) immunodetection system (WP20005, Thermo Fisher Scientific Inc.). Membranes were cut to allow simultaneous incubation of different primary antibodies on the same samples. For the preparation of figures, we cropped the original Western blot to generate the appropriate figure panels with the relevant lanes. The cropped image was then subjected to uniform image enhancement of contrast and brightness. Molecular weights were determined using the SeeBlue™ Plus2 Pre-stained Protein Standard (Invitrogen, Thermo Fisher Scientific Inc.).

### 4.6. ELISA

Periostin and sparc plasma concentrations were assessed in 100 plasma samples from AS patients and 21 plasma samples from clinically relevant PCa patients by using ELISA assays, according to the manufacturer’s instructions. The following tests were used: Human Periostin kit (EHPHSTN, Invitrogen, Thermo Fischer Scientific Inc.), Human SPARC Quantikine ELISA (DSP00, R&D System). To obtain a plasma sample, whole blood was collected in anticoagulant tubes and plasma was immediately separated by double centrifugation at 2200× *g* for 20 min and further clarified at 2200× *g* for 10 min. The colored product generated by the enzyme activity was detected by plate-reader POLARstar optima (VWR International, Radnor, PA, USA) at 450 nm.

### 4.7. Statistical Analyses

In the analysis of in-vitro experiments, the test of Mann–Whitney was used to assess differences in continuous variables between groups of interest.

In the analysis of the FFPE prostatectomy sample cohort, the association of periostin with each of the other considered variables (resection margins, prognostic grade group, seminal vesicle invasion, extraprostatic extension, and cribriform morphology) was investigated using a univariate logistic regression analysis, Chi-square statistic or Fisher’s exact test whenever appropriate.

Biochemical recurrence (BCR)-free survival was calculated as the time from surgery to the first biochemical failure, defined by a PSA ≥ 0.2 ng/mL. The patterns of BCR-free survival were estimated using the Kaplan–Meier method [28], and the survival curves were compared using log-rank tests by considering the periostin staining score dichotomized as ‘low’ if equal to 0 or 1 and ‘high’ if equal to 2 or 3. The prognostic role of each considered variable on BCR-free survival was investigated using a Cox regression analysis in both univariate and multivariate models [29].

The initial multivariate model included all of the variables that were statistically significant (alpha = 5%) at univariate analysis and not associated with the periostin variable. A more parsimonious final model was then obtained using a backward selection procedure that retained only those variables reaching the conventional level of significance of 5%. In any case, to avoid losing important prognostic biomolecular markers, regardless of significant association with periostin, each of those variables previously rejected was investigated by individually adding them to the final model. Finally, we evaluated the predictive capability of the final model and the contribution of each variable using Harrell’s C-statistics [30]. Statistical analyses were performed with the SAS software (Version 9.4; SAS Institute Inc., Cary, NC, USA) by adopting a significance level of alpha = 5%.

In the analysis of the plasma sample cohort, descriptive statistics included a mean, median, and interquartile range for numerical variables and frequency tables for categorical variables. The distribution of numerical variables among groups was presented graphically using box plots, while the differences were tested statistically using the Wilcoxon test. The association of the covariates of interest to the group was estimated using univariate and multivariate logistic regressions. The model’s performance was evaluated through Harrell’s C-statistics and the area under the ROC curve, the 95% confidence interval of which was estimated with bootstrap [29]. Continuous numerical variables were modeled using restricted cubic splines with three knots positioned at the quartiles. An index score was constructed based on the biomarkers of interest using the methodology of Adapted Index Modelling for binary outcomes [31]. The methodology allows for the categorization of the numerical variables by estimating the optimal cut-off with respect to the outcome. In this way, patients are grouped into high- and low-risk strata for each variable. Each time a patient is found in the high-risk group, a point is assigned. The final index score is obtained as the sum of single indicators for each variable, ranging thus from 0 to the number of variables included in the index. Higher scores are indicative of higher risk.

## Figures and Tables

**Figure 1 ijms-23-07987-f001:**
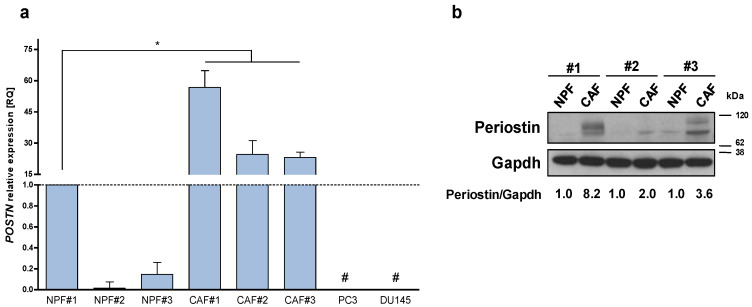
Periostin is over-expressed in CAFs. (**a**) Expression levels of *POSTN* mRNA were assessed by qRT-PCR in a panel of three patient-derived CAFs isolated from the tumoral area of radical prostatectomy samples and matched normal prostate fibroblasts (NPFs) isolated from non-tumoral areas and PCa cell lines (DU145, PC3). Data represented as mean ± SD of three independent experiments with respect to NPF#1 (Mann–Whitney one-sided test, * = *p* < 0.05). # = undetected values. Ct value of POSTN in NPF#1 is 27. (**b**) Western blotting analysis showing periostin protein levels in three paired CAFs and NPFs. Quantification, as indicated under the blot, was estimated from the intensity of Western blot signals using the ImageJ software. The periostin/gapdh ratios were relativized to 1 for each paired sample.

**Figure 2 ijms-23-07987-f002:**
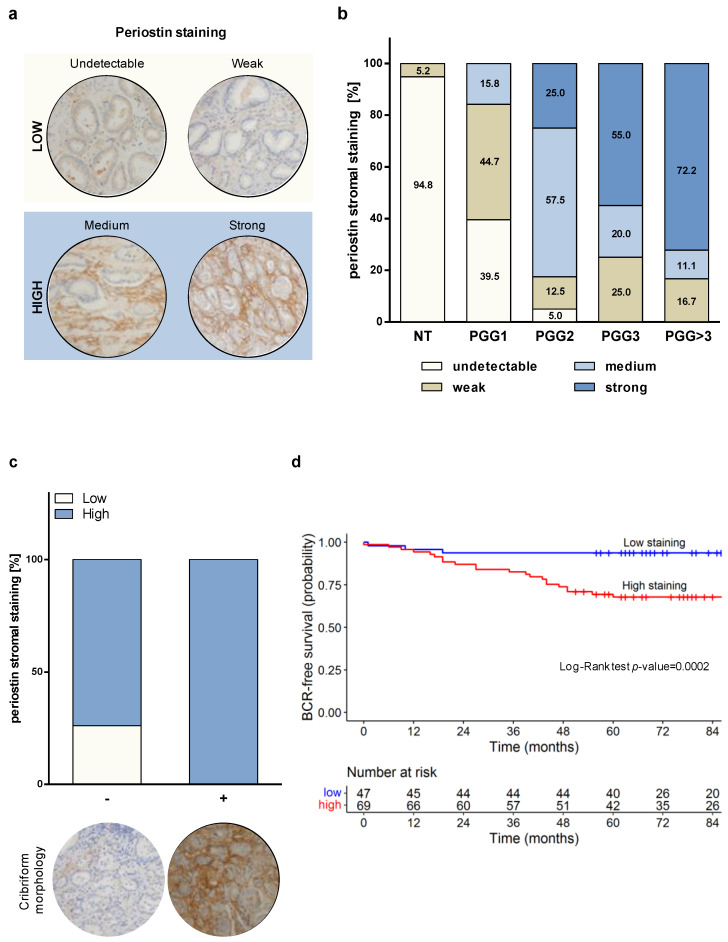
Periostin stromal staining correlates with tumor aggressiveness and biochemical relapse in PCa patients. (**a**) Immunohistochemical scoring of periostin in representative PCa samples: undetectable and weak score staining were assigned to low category; medium and strong score staining were included in the high category. Periostin staining was mainly observed in peritumoral areas. Magnification: 200×. (**b**) Stromal periostin score distribution within prognostic grade groups (PGG) of 116 PCa samples. (**c**) Periostin stroma staining distribution in 60 GS = 7 (PGG2/3) tumors, classified based on the presence (+) or absence (-) of cribriform morphology. (**d**) BCR-free survival of 116 PCa patients categorized based on stromal periostin score. In the corresponding table, the number of patients at risk at the different time points for each of the two groups (low and high staining) is reported.

**Figure 3 ijms-23-07987-f003:**
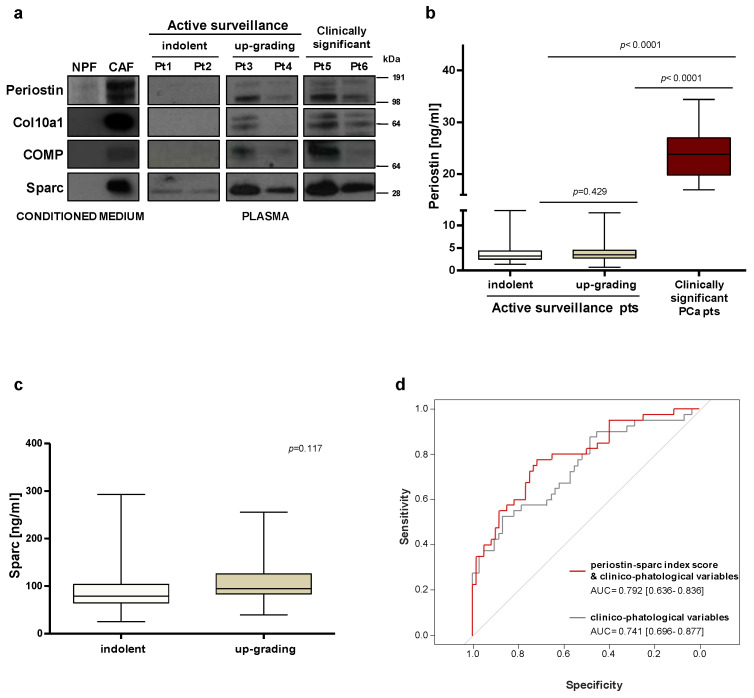
Circulating levels of periostin and sparc may be used to improve risk stratification. (**a**) Protein expression levels of selected stromal proteins were assessed by Western blotting in the conditioned medium from CAFs and NPFs and in plasma samples collected from PCa patients. (**b**) ELISA–based evaluation of circulating periostin in plasma samples collected at the baseline from 100 AS patients, including 40 who discontinued the program upon upgrading after one year (up–grading group) and 60 who remained in AS for more than 5 years (indolent group), and in 21 patients with clinically significant PCa. The Wilcoxon test was performed and *p* < 0.05 was considered significant. (**c**) ELISA–based evaluation of circulating sparc in plasma samples collected at the baseline from 100 AS patients, including 40 who discontinued the program upon upgrading after one year (up-grading group) and 60 who remained in AS for more than 5 years (indolent group). The Wilcoxon test was performed and *p* < 0.05 was considered significant. (**d**) ROC curves and corresponding AUCs derived from the index score models built on clinico–pathological variables only and upon addition of circulating sparc and periostin in 100 AS patients.

**Table 1 ijms-23-07987-t001:** Results from the univariate and multivariate Cox analysis on the prostatectomy cohort.

	Univariate Analysis		Multivariate Analysis	
Variables	HR	95% CI	HR	95% CI
Periostin				
2–3	6.98	2.11–22.95	5.75	1.73–19.17
0–1a	-		-	
Resection margins				
>0	3.83	1.77–8.29	3.47	1.6–7.50
0a	-		-	
PGG				
>2	3.93	1.46–10.61		
2	1.42	0.45–4.47		
1a	-			
Seminal vesicle invasion				
pos	3.57	1.51–8.47		
neg ^a^	-			
EPE				
pos	3.38	1.54–7.39		
neg ^a^	-			

^a^ Reference category; Periostin: periostin stromal staining (dichotomized as high and low); Resection margins (dichotomized as positive and negative); PGG: prognostic grade group (classified as PGG1, PGG2, and PGG > 2); Seminal vesicle invasion (dichotomized as positive and negative); EPE: extraprostatic extension (dichotomized as positive and negative); HR: Hazard ratio; CI: Confidence Interval.

## Data Availability

The data set used and analyzed during the current study are available for research purpose.

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
