# Peer review of "Potential of the Stromal Matricellular Protein Periostin as a Biomarker to Improve Risk Assessment in Prostate Cancer"

_ijms, 2022, doi:10.3390/ijms23147987_

Round 1

Reviewer 1 Report

The authors describe the potential of a stromal protein periostin as a biomarker. The idea is clear, it is important to be able to stratify patients at risk of development of aggressive PCa. The use of the tumor supportive stroma in stead of the markers found in the tumor is a well known idea that would be of benefit especially in minimal residual disease when there is a very low tumor mass but potentially more tumor-associated stroma with respective biomarkers. 

My main concern with this paper is the use of only one/two biomarker(s) to stratify patients at risk of progressing. This is not very convincing and will probably not proceed into clinical practice, especially since the authors did not show significant differences in plasma expression levels between AS patients in the indolent group vs the up-grading AS group. (only vs the clinically significant PCa) 

Moreover, this paper neglects a publication regarding a 7-gene signature (of which one is periostin) for osteoblastic-specific bone metastastic stromal gene expression. (Ozdemir et al Plos One 2014). In this paper, the expression of periostin is described in the stroma of bone metastatic human PCa but also in primary prostate cancer (as well as in bone metastatic mammary carcinoma). This paper should be included and the stromal expression of periostin described in this paper should be included in introduction and discussion section. 

Result section:

Figure 1: The authors describe gene expression levels un CAFs vs NAFs in 1a. Since these cells are cultured first on tissue culture plastic, this could severely affect gene expression.  What is the Ct value of the NPF#1? This should be mentioned, since it is important for the interpretation of the results (do we see a 60x increase of "almost no expression"?)

It is not clear whether the patient derived CAFs and NAFs (1b) were also cultured before assessing protein expression levels. (see above comment on expression levels when cultured on tissue culture plastic

Western Blot data figure 1b and Figure 3a: in figure 1 you can clearly see 2 bands with the periostin protein. this is not visible due to overexposure in the conditioned medium experiment in figure 3. These picture is quite dark, making it difficult to see the bands clearly. Moreover in figure 3, the loading control is missing (GAPDH?). In both WB pictures, molecular weight marker is missing.

How do the authors explain the discrepancy between the periostin levels found in the western blot vs the ELISA data? Is it just coincidence you found higher level in one of the 2 AS up-grading patients? Was this checked on more patients? Did the amount of periostin protein detected in the western blot correlate with the ELISA-based data?

Figure 3d. Has the periostin expression been validated in another (small) cohort of patients?

Discussion section:

Sentence 256 We provide evidence that circulating stromal biomarkers can improve the performance for predicting disease reclassification: this evidence is not convincingly provided in the manuscript. No significant changes were found between AS indolent and AS upgrading group. Significant changes were only found between AS and clinically significant disease. 

In the discussion, several papers were discussed stating that relevant studies in PCa missed the identity of periostin expressing cells within the tumor mass. However, this was also not thoroughly studied in this manuscript. No co-stainings with known stromal cell markers were performed in the clinical specimens neither in the cell cultures. How did the authors ensure the isolated cells are NAFs / CAFs and not other stromal cells or PCa tumor cells? Which markers were used? 

Reviewer 2 Report

The authors explored an important issue in prostate cancer diagnosis: how to better monitor the low-risk prostate cancer patients and predict aggressive prostate cancer. They discovered that periostin and sparc in the blood can help refine the risk assessment and better predict cancer progression for patients under active surveillance. Overall, the manuscript was well-written. The authors may use less jargons for the benefits of readers with different backgrounds. The authors provided many data to support their conclusions. The critical comments are listed below:

11)     The author may consider change some of the jargons and use less abbreviations for the readers’ benefit. For example, BCR, NPFs, CAFs. What is GSE76260 in line 110?

22)     In Fig. 1b, densitometry of the Western blots should be included.

33)     In Fig. 2d, it’s not clear what the bottom table is related to the top blue/red line graph. Please explain more

4) Line 247 to 249 was probably included erroneously. “explanation” should be “explantation” in line 386.
